# CLASS INCREMENTAL CONTINUAL LEARNING WITH SELF-ORGANIZING MAPS AND SYNTHETIC REPLAY

## ABSTRACT

This work introduces a novel generative continual learning framework based on self-organizing maps (SOMs) extended with learned distributional statistics and encoder–decoder models which enable memory-efficient replay, eliminating the need to store raw data samples or task labels. For high-dimensional input spaces, the SOM operates over the latent space of the encoder–decoder, whereas, for lower-dimensional inputs, the SOM operates in a standalone fashion. Our method stores a running mean, variance, and covariance for each SOM unit, from which synthetic samples are then generated during future learning iterations. For the encoder–decoder method, generated samples are then fed through the decoder to then be used in subsequent replay. Experimental results on standard class-incremental benchmarks show that our approach performs competitively with state-of-the-art memory-based methods and outperforms memory-free methods, notably improving over the best state-of-the-art single class incremental performance without pretrained encoders on CIFAR-10 and CIFAR-100 by nearly 10% and 7%, respectively. We also find best performance on single class incremental CIFAR-100 utilizing a foundational encoder–decoder, and present the first baseline results for single class incremental TinyImageNet. Our methodology facilitates easy visualization of the learning process and can also be utilized as a generative model post-training. Results show our method's capability as a scalable, task-label-free, and memory-efficient solution for continual learning.

## 1 INTRODUCTION

Computational systems deployed in real-world environments are often exposed to continuous streams of information, where the data distribution(s) that the system receives change over time. In such environments, the machine learning models that are set up must adapt to new tasks sequentially, without revisiting previous data, while ensuring that they retain knowledge extracted from previous tasks. The ability of systems to learn new tasks, while retaining knowledge of past experiences, is referred to as continual learning (CL), or lifelong learning (Thrun, 1998), and is central to building robust intelligent systems.

Continual learning research typically considers three main scenarios: task-incremental (TIL), domain-incremental (DIL), and class-incremental (CIL) (van de Ven et al., 2022). In TIL, task identity is provided at inference, making it the easiest setting. DIL removes task identity but keeps the same class set across domains. CIL is the most challenging, as task identity is unknown and the model must discriminate among all classes seen so far while being exposed to only a subset at a time. The extreme case is single-class CIL, where data arrives one class at a time. Prior surveys on continual learning note that many benchmarks are defined in a multi-class-per-task fashion (e.g., Split-MNIST, Split-CIFAR), where each task introduces several new classes simultaneously (Yang et al., 2025; Zhou et al., 2024a; Wickramasinghe et al., 2024). In contrast, this work focuses on the single-class-incremental learning setting, widely recognized as the most challenging protocol (Maltoni & Lomonaco, 2019), since the model must incrementally separate classes without ever jointly observing them.

The challenges of CL become particularly evident in the context of deep neural networks (DNNs). While DNNs have achieved remarkable success across vision, language, and reinforcement learning tasks (Samek et al., 2021; Doon et al., 2018; Ying et al., 2024), they are highly susceptible to

catastrophic forgetting (Parisi et al., 2019; McCloskey & Cohen, 1989; Ororbia et al., 2022) when trained on sequential, non-i.i.d. data streams. Several DNN-based approaches have been developed in the field of CL (Wang et al., 2024); however, most focus on supervised methods, which are often difficult to interpret due to model complexity. To better explore CL more clearly, some studies have turned to unsupervised methods (Ashfahani & Pratama, 2023; Madaan et al., 2022; Hirani et al., 2024; Ororbia, 2021). These methods aim to learn evolving data distributions while maintaining previously acquired representations, often by leveraging latent space structure or clustering dynamics.

While extensive research has been done with DNNs in the area of CL (Zhou et al., 2024b), self-organizing maps (SOMs) (Kohonen, 1990), a class of unsupervised, topology-preserving neural models, have received relatively little attention despite their natural suitability for such settings. SOMs consist of a number of unit vectors, and, during training, only the best matching unit and its neighbors are updated in response to an input. This localized plasticity can help preserve previously learned representations and prevent forgetting; and this has lead to several studies of their use in CL (Ororbia, 2021). Recent work has explored combining SOMs with neural architectures to improve continual learning – SOMLP (Bashivan et al., 2019) utilizes a SOM layer to gate an MLP's hidden units to reduce forgetting without requiring memory buffers or task labels. More recently, the dendritic SOM (DendSOM) (Pinitas et al., 2021) utilizes multiple localized SOMs to mimic dendritic processing, enabling sparse, task-specific learning. A related approach, the continual SOM (c-SOM) (Vaidya et al., 2021), introduces internal Gaussian replay in the input space, but the absence of a proper generative model limits its scalability and sample diversity.

This paper proposes a novel family of unsupervised continual learning models that integrates extended Self-Organizing Maps (SOMs) with encoder–decoder architectures (for this work, convolutional VAEs (Kingma & Welling, 2013) and the foundational CLIP model (Radford et al., 2021)) to address catastrophic forgetting in class-incremental settings. Main contributions of this paper are:

- A novel SOM-based CL framework that combines generative modeling with topology-preserving clustering for easily visualized unsupervised class-incremental learning.
- Three variants of the framework: *(i)* a SOM-only for low-dimensional data, *(ii)* a global encoder–decoder model jointly trained with the SOM to provide structured latent spaces, and *(iii)* per-BMU specialized encoder–decoder models for fine-grained generative replay.
- A memory-efficient generative replay scheme that stores only summary statistics (mean, variance, and covariance) per SOM unit, avoiding external buffers or replay data.
- Extensive evaluation in both non-pretrained and pretrained settings on MNIST, CIFAR-10, CIFAR-100, and TinyImageNet. Experiments demonstrate strong knowledge retention and scalability across increasing data complexity, showing state-of-the-art results in both single and multi class incremental settings.

## 2 METHODOLOGY

SOMS (also known as Kohonen maps) (Kohonen, 1990) offer a unique approach to unsupervised learning by mapping complex, multidimensional data into a two-dimensional grid. The strength of SOMs lies in their ability to capture the high-dimensional variance of data and represent it on a grid that is visually interpretable. SOMs contain a grid of unit vectors, and during training sample vectors are mapped to their best matching unit (BMU), which pulls the BMU and other unit vectors within a neighborhood radius towards the sample – with unit vectors farther away from the BMU having smaller updates. Once the SOM is trained, an input vector can be assigned to its BMU, which serves as a representative anchor point for that input in the topological map. In the context of labeled datasets, each BMU can accumulate label distributions support downstream interpretability or weakly supervised clustering.

We propose a class-incremental continual learning framework that utilizes self-organizing maps (SOM) extended with learned per-unit distributions (running mean, variance and covariance), shown in Algorithm 1. The SOM is trained with raw or embedding-level samples, and the per-unit summary statistics are utilized for generative replay – either of raw data or by passing generated embeddings into a decoder – allowing the SOM units to play a central role in learning by acting as generative memory units. This approach is capable of adapting to the complexity of the input data: simple grayscale images (e.g., MNIST) can be processed using raw data and the SOM alone, whereas

higher-dimensional RGB images (e.g., CIFAR-10/100, TinyImageNet) can utilize embeddings from a larger scale model such as a variational autoencoder or foundation model for a more efficient and structured representation. Note this method is entirely unsupervised as class labels are not used to drive the weight updates in the SOM or embedding models. Class labels are tracked on BMU matches during training only so they can be used for testing the performance of using the SOM for inference. As labels are not required, this allows our methodology to be used for single class incremental learning, unlike many other continual learning models (Yang et al., 2025; Wang et al., 2024; Wickramasinghe et al., 2024). For evaluation, we use the final, trained SOM output as a classifier, where each unit is assigned a class label based on the most frequent label among the samples mapped to it during training, i.e., majority voting based on best matching unit (BMU) "hit" counts.

## 2.1 Tracking SOM Unit Distribution Statistics

For an $n \times n$ SOM grid and a momentum factor $\alpha$, each input $x$ is projected to its corresponding BMU, which is used to update the BMU's core properties: i), a running **mean vector**: $\mu_{ij} \leftarrow (1-\alpha)\mu_{ij} + \alpha x$; ii), a running **variance vector** $\sigma_{ij}^2$, capturing per-dimension variability, calculated by: $\sigma_{ij}^2 \leftarrow (1-\alpha)\sigma_{ij}^2 + \alpha(\mu_{ij} - x)^2$; and iii), a running **covariance matrix** $\Sigma_{ij}$, which helps in modeling inter-feature relationships, calculated by: $\Sigma_{ij} \leftarrow (1-\alpha)\Sigma_{ij} + \alpha(x - \mu_{ij})(x - \mu_{ij})^\top$. Here, $x$ denotes a sample pattern vector when the SOM is standalone, otherwise, it is the latent vector encoding of the input, and is assigned to the BMU at position $(i, j)$ on the SOM grid. These running statistics characterize the local distribution of latent codes associated with each BMU.

However, a practical issue with these running statistics is that they are biased toward their initialization in the early stages of training. For example, when updating the running mean of a BMU at location $(i, j)$ as $\mu_{ij,t} = \alpha\mu_{ij,t-1} + (1-\alpha)x_t$, when initialized as $\mu_{ij,0} = 0$, the estimate $\mu_{ij,t}$ underestimates the true mean since it is implicitly influenced by the zero initialization. This effect is especially problematic in continual learning settings, where some BMUs may receive very few samples early on. To mitigate this, we employ *bias correction* in the style of the Adam optimizer (Kingma & Ba, 2017). Specifically, the corrected estimates for each BMU $(i, j)$ are:

$$\hat{\mu}_{ij,t} = \frac{\mu_{ij,t}}{1 - \beta_\mu^t}, \quad \hat{\sigma}_{ij,t}^2 = \frac{\sigma_{ij,t}^2}{1 - \beta_\sigma^t}, \quad \hat{\Sigma}_{ij,t} = \frac{\Sigma_{ij,t}}{1 - \beta_\Sigma^t},$$

where $t$ denotes the number of update steps (BMU matches) received by the BMU, and $\beta_\mu, \beta_\sigma, \beta_\Sigma \in [0, 1)$ are the statistic specific exponential decay rates. This bias correction ensures that BMU parameters $(\mu_{ij}, \sigma_{ij}^2, \Sigma_{ij})$ remain unbiased from the start, yielding stable local distribution estimates and well-conditioned covariances for synthetic sampling. The neighboring units are updated with decayed learning rates based on their distance to the BMU, preserving SOM topology.

## 2.2 Continual Learning and SOM Unit Synthetic Replay

To utilize this method for class incremental learning, our model is trained sequentially as new classes arrive. At task $t$, only the data from class $C_t$ is available. To retain knowledge from previous classes $C_0, C_1, \ldots, C_{t-1}$, our method generates synthetic samples using these distributional statistics, and replays them alongside new class data. For multi-class incremental learning, more than one class can be provided per task. The unit distribution statistics allow the SOM to serve as a class- and region-specific memory that accumulates information over time, mitigating forgetting. When a new task $t$ consisting of a disjoint subset of classes is introduced, synthetic data from previous classes is generated by sampling from the stored Gaussian distributions of their corresponding BMUs:

$$\tilde{x} \sim \mathcal{N}(\hat{\mu}_{ij}, \hat{\sigma}_{ij}^2) \quad \text{(for MNIST)}$$

$$\tilde{z} \sim \mathcal{N}(\hat{\mu}_{ij}, \hat{\Sigma}_{ij}) \quad \text{(for CIFAR-10/CIFAR-100/TinyImageNet)}.$$

It should be noted that, when using the covariance method to generate the samples from the distribution, we need to apply eigenvalue regularization to ensure numerically stable sampling (Algorithm 2, Appendix A). This procedure eliminates negative or near-zero variance directions that may otherwise lead to instability during sampling and ensures that the multivariate normal distribution remains valid and well-conditioned.

While we use boundaries to organize the training phases (for experimental simulation) and trigger the generation of synthetic samples from previously seen classes, the model itself is trained without access to task IDs or class labels. Replay is scheduled externally, but the CLIP, VAE and SOM modules update solely based on input data, making no distinction between real and replayed samples. This places our method in a more challenging CL context, closely aligned with task-free CL (Lee et al., 2020; Ororbia, 2021), where explicit supervision about task transitions is not available during model updates.

Note that, unlike classical buffer-based replay, our SOM system design leverages only these statistical summaries stored at each BMU. This enables highly compact memory usage as it scales with respect to the fixed SOM grid size as opposed to the dataset size. Figure 1 shows synthetic samples generated from an SOM fully trained, class-incrementally, on CIFAR-100, where each latent vector was sampled from the full-covariance Gaussian

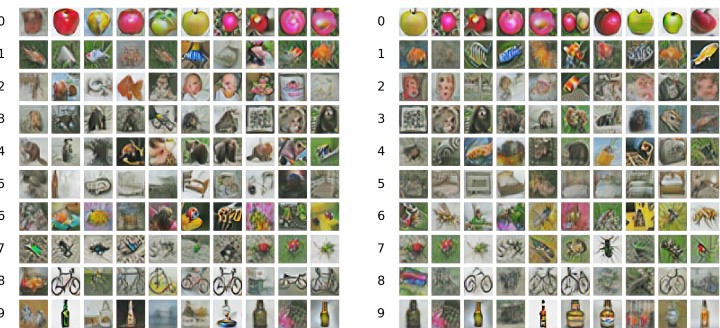

Figure 1: Synthetic CIFAR-100 samples for the first 10 classes generated by sampling latent vectors from an SOM trained on VAE latent space (left) and CLIP embeddings (right). Each row corresponds to one class (0–9).

distribution of a BMU associated with the target class and subsequently decoded by the VAE. For experiments that use CLIP embeddings, we employ a decoder trained on CLIP embeddings to reconstruct images from the sampled latent vectors. Appendix C provides similar visualizations of images generated from the SOM unit vectors for other datasets. Over all datasets, we find that the SOM systems generate feasible variations of their learned class.

Our methodology also provides interpretability via the SOM grid's topological visualization of class structure and supports modularity, since synthetic replay depends solely on local BMU statistics. This makes it possible to easily view the progress of the SOM by decoding unit vectors after each task to visually see how the model is performing (see Appendix E). If the method is performing well, classes have well formed clusters within the SOM.

## 2.3 HIGH-DIMENSIONAL REPLAY CHALLENGES AND THE ROLE OF LATENT COMPRESSION

For low-dimensional image datasets such as MNIST, synthetic sample generation using SOMs is generally efficient and effective due to the low dimensionality of the feature space ($28 \times 28$ grayscale pixels), where the variance along each independent dimension is sufficient to model the data distribution. However, this mean-variance sampling strategy is insufficient when applied to high-dimensional datasets, e.g., CIFAR-10 and CIFAR-100; and with preliminary results for sample generation using the mean-variance method above were of poor quality and degraded classification performance. Additionally, using the mean covariance method in pixel space introduces memory storage issues, e.g., CIFAR-10/100 requires storing a full $3072 \times 3072$ covariance matrix for each BMU.

To address this, we allow the use of embeddings from any generative model, which serve to compress high-dimensional images into a compact latent space. In this work, we investigate two instantiations: a non-pretrained convolutional VAE (Kingma & Welling, 2013; 2019) and a pretrained foundation model, CLIP (ViT-B/32) (Radford et al., 2021). This significantly reduces the dimensionality of each SOM BMU; for example, the CLIP encoder produces a 512-dimensional embedding, while our VAE configuration uses a $d$-dimensional latent space (e.g., $d = 128$). Such compression makes it feasible to model and store full covariance matrices. By sampling from the full multivariate Gaussian $\mathcal{N}(\mu, \Sigma)$ in this latent space, we generate high-quality synthetic representations by feeding these synthetic samples through the models decoder, replaying them in subsequent training phases. This hybrid approach ensures scalable, efficient, and expressive sample replay for complex image distributions in CL settings. It is also flexible in that different encoder/decoder models can be substituted

**Algorithm 1** Unified Algorithm for Class-Incremental Learning with SOM and Optional Encoder/Decoder Models (e.g., VAE, CLIP)

**Input**: Dataset $\mathcal{D} = \{\mathcal{D}_c\}_{c=0}^{C-1}$; flags USE_GLOBAL_ENCDEC, USE_PER_BMU_ENCDEC; replay samples per BMU $K$
**Output**: SOM, (optional) global encoder/decoder, (optional) per-BMU encoders/decoders $\mathcal{V}$

```
1:  Initialize SOM; initialize replay buffer R ← ∅
2:  if USE_GLOBAL_ENCDEC then initialize global encoder/decoder
3:  if USE_PER_BMU_ENCDEC then initialize per-BMU model dictionary V ← ∅
4:  for c = 0 to C−1 do
5:      T_c ← D_c if c=0 else D_c ∪ R
6:      // Encode features for SOM update
7:      if USE_GLOBAL_ENCDEC then
8:          Train global encoder/decoder on T_c
9:          Z_c ← Enc_global(T_c)
10:         Update SOM with Z_c (including neighborhood updates)
11:     else
12:         Update SOM with T_c (including neighborhood updates)
13:     end if
14:     // Train per-BMU encoders/decoders on assigned subsets
15:     if USE_PER_BMU_ENCDEC then
16:         Assign each x ∈ T_c to BMU (i, j) using its current representation (Z_c if global encoder used, else raw T_c)
17:         for each BMU (i, j) with assigned set S_ij ⊆ T_c do
18:             Train or update local model M_ij on S_ij
19:             Store/refresh V[(i, j)] ← M_ij
20:             Re-encode S_ij with M_ij encoder to refine SOM neighborhood updates
21:         end for
22:     end if
23:     // Build replay for next class
24:     R_c ← ∅
25:     for each BMU (i, j) do
26:         Obtain BMU stats (μ_ij, Σ_ij) (from running latent/feature stats at (i, j))
27:         (μ̂_ij, σ̂²_ij, Σ̂_ij) ← BIASCORRECTION(μ_ij, σ²_ij, Σ_ij, β, t_ij, λ)
28:         for k = 1 to K do
29:             z̃ ∼ N(μ̂_ij, Σ̂_ij)
30:             x̃ ← { Dec_ij(z̃),   USE_PER_BMU_ENCDEC & (i, j) ∈ V
                      Dec_g(z̃),    USE_GLOBAL_ENCDEC
                      z̃,           SOM-only
31:             Append x̃ to R_c
32:         end for
33:     end for
34:     Set R ← R_c
35: end for
36: return SOM, (global encoder/decoder if used), (per-BMU models V if used)
```

depending on the dataset and experimental requirements. This setup emphasizes that our contribution lies not in the specific encoder–decoder, but in the replay framework itself. Whether the latent codes or embeddings originate from a VAE or a large pretrained transformer, the SOM provides the same modular structure for clustering, memory-efficient replay, and continual adaptation.

**VAE Instantiation.** The VAE encoder maps inputs into a structured latent distribution parameterized by mean $\mu$ and log-variance $\log \sigma^2$, with sampling performed via the reparameterization trick: $z = \mu + \sigma \odot \epsilon$, $\epsilon \sim \mathcal{N}(0, I)$. The VAE decoder reconstructs the image from $z$, enabling generative replay by reconstructing synthetic samples drawn from SOM statistics (means and covariances). The encoder and decoder are implemented using residual downsampling and upsampling blocks (He et al., 2016), providing stable and expressive nonlinear transformations. Depending on the configuration, the VAE compresses images into latent vectors of dimension $d$ (e.g., $d = 128$), achieving over 90% dimensionality reduction compared to raw pixel space.

**CLIP Instantiation.** To demonstrate the generality of the framework, we also evaluate our replay strategy using CLIP (ViT-B/32) (Radford et al., 2021), a large vision–language foundation model. Instead of training an encoder from scratch, we extract 512-dimensional embeddings from CLIP's penultimate transformer block. The CLIP encoder are either kept frozen or fine-tuned, depending on the experimental setting. Once the embeddings are extracted, they are fed directly into the SOM, and the replay mechanism is identical to the VAE case, where the SOM maintains class-specific statistics over the embeddings and uses them to generate synthetic embeddings. These sampled embeddings are then passed through a decoder trained on CLIP embeddings to reconstruct images for replay. The full decoder architecture used for CLIP-based reconstruction is provided in Appendix 12.

**Perceptual Quality via Feature Loss.** To improve reconstruction quality during replay, we add a perceptual feature loss in addition to pixel-level and KL terms. Given real images $x$ and reconstructions $\hat{x}$, the loss is

$$\mathcal{L}_{\text{feat}} = \sum_{\ell=1}^{L} \|\phi_\ell(x) - \phi_\ell(\hat{x})\|_2^2 \,, \tag{1}$$

where $\phi_\ell$ are activations from a frozen pretrained feature extractor. For VAEs, we use VGG-19 (Simonyan & Zisserman, 2015); for CLIP-based replay, we instead use the frozen CLIP encoder (Radford et al., 2021). This unified formulation ensures reconstructions preserve high-level semantics while the SOM maintains consistent latent representations.

The encoder-decoder framework, as described above, provides several advantages in CL by reducing input dimensionality, enabling efficient memory usage for SOM BMU statistics, and supporting fast, realistic synthetic replay for SOMs. It acts as a front-end compression module, transforming high-dimensional images into a structured, low-dimensional latent space. This ultimately makes the overall SOM-based system training faster and allows for reliable Gaussian-based sample generation to induce replay.

### 2.4 Localized Replay with BMU-Specific Encoder–Decoders

As an extension to our core CL framework, we explored a modular generative replay strategy wherein a separate model is trained for each SOM BMU (Algorithm 1). While a single global encoder–decoder (e.g., VAE) combined with the SOM is efficient and benefits from training across all data, its decoder must generalize over a wide variety of samples, including those that may be underrepresented in the global latent space. In contrast, the per-BMU approach assigns a dedicated local model to each SOM unit, at the cost of additional memory. After training the global encoder and SOM, each input is mapped to its BMU, and the original images associated with that unit are used to train its local model. Once trained, the encoder part of the local model provides refined latent representations that can be used to update SOM weights for the BMU and its neighbors, while the decoder specializes in reconstructing samples from that region of the latent space. This results in a collection of localized decoders aligned with the topological structure of the SOM. During replay, synthetic latent vectors are sampled from the SOM's bias-corrected statistics at the BMU, and decoded using the corresponding local model rather than the shared global decoder. The motivation behind this variation is to align generative capacity with the SOM's topology, so each decoder specializes in a region of the latent space—leading to improved reconstructions.

## 3 Results

Our methodology was evaluated across widely-adopted CL benchmarks: MNIST, CIFAR-10, CIFAR-100 and TinyImageNet, first in the more challenging single class incremental setting, where classes are presented one at a time (that is, class 0, then class 1, etc.). We also evaluated our methodology on standard split versions of these datasets, set up in a task-incremental fashion, where each task contains $N$ disjoint classes, e.g., for a task size of two the first task would have classes 0-1, the second 2-3, etc. Results utilize the best found training and initialization hyperparameters, which were taken after significant ablation studies (see Appendices A and B).

### 3.1 One Class Per Task Incremental Learning

Table 1 compares our methodology with several well-known CL methods for the single class incremental setting (MNIST and CIFAR 10 have 10 tasks, CIFAR-100 has 100 tasks, and TinyImageNet has 200 tasks; one class per task for each). We compare pretrained and non-pretrained methodologies, bias and non-bias correction, and our three methods for incorporating generative models – a global fixed model (frozen, global), a global model that is trained concurrently with the SOM (FT global), and models concurrently trained for each SOM unit (FT BMU specific). For MNIST, our SOM-only approach achieves an accuracy of 95.16% without the bias correction, closely matching the best performing rehearsal-based methods, including DisCOIL (96.69%) and PCL (95.75%). With bias correction, the accuracy increased to 95.88%. Traditional regularization-based approaches such as EWC and LwF perform poorly on CIFAR-10 (10.01%, 10.05%) and CIFAR-100 (1.03%,

Table 1: Final classification accuracy for single class incremental learning.

| Method | w/o pretraining | | | w/ pretraining | | TinyImageNet |
|---|---|---|---|---|---|---|
| | MNIST | CIFAR10 | CIFAR100 | CIFAR10 | CIFAR100 | |
| EWC | 9.91 | 10.01 | 1.03 | 10.21 | 2.93 | – |
| LwF | 19.96 | 10.05 | 2.13 | 19.39 | 6.25 | – |
| IMM | 29.16 | 10.25 | 1.21 | 51.22 | 12.58 | – |
| PGMA | 71.36 | 20.08 | 1.86 | 56.22 | 12.37 | – |
| RPSNet | 40.29 | 16.31 | 1.96 | 55.54 | 4.13 | – |
| OWM | 94.46 | 19.63 | 3.67 | 83.03 | 63.26 | – |
| PCL | 95.75 | 31.58 | 5.58 | **84.93** | 63.61 | – |
| DisCOIL | **96.69** | 44.54 | – | – | – | – |
| PCL-L2 | – | – | – | 77.95 | 54.83 | – |
| **Ours (SOM-based w/o bias)** | | | | | | |
| SOM only | 95.16 | – | – | – | – | – |
| VAE (FT global) | 93.22 | 54.16 | 12.41 | – | – | 7.14 |
| VAE (FT BMU specific) | 92.85 | 46.10 | 12.15 | – | – | 6.45 |
| SOM+CLIP (frozen, global) | – | – | – | 78.12 | 61.22 | 40.99 |
| SOM+CLIP (FT global) | – | – | – | 81.22 | 63.12 | 41.67 |
| SOM+CLIP (FT BMU specific) | – | – | – | 83.11 | **63.78** | 43.11 |
| **Ours (SOM-based w bias)** | | | | | | |
| SOM only | 95.88 | – | – | – | – | – |
| VAE (FT global) | 93.26 | **54.58** | **12.66** | – | – | 7.66 |
| VAE (FT BMU specific) | 92.97 | 49.22 | 12.18 | – | – | 6.78 |
| SOM+CLIP (frozen, global) | – | – | – | 79.11 | 62.65 | 42.48 |
| SOM+CLIP (FT global) | – | – | – | 81.34 | 63.22 | 43.26 |
| SOM+CLIP (FT BMU specific) | – | – | – | 83.56 | **64.88** | **45.11** |

2.13%), indicating significant forgetting in the one-class stream. More advanced strategies like PGMA (Hu et al., 2019), RPSNet (Rajasegaran et al., 2020), and OWM (Zeng et al., 2018) achieve lower gains, with CIFAR-100 scores ranging from 1.86% to 3.67%. The memory-based approaches, such as PCL (Hu et al., 2021) and DisCOIL (Sun et al., 2022), perform significantly better (up to 44.54% in CIFAR-10), although they often rely on external memory/labels. In contrast, our method (with bias correction) achieves 54.58% in CIFAR-10 and 12.66% in CIFAR-100 without access to external exemplars or task identifiers, outperforming most baselines by large margins. Notably, the VAE (FT BMU specific) variant surpasses previous methods on CIFAR-100 (12.15% without bias correction; 12.18% with bias correction), highlighting the effectiveness of our generative replay strategy based on structured topological organization. With CLIP embeddings, SOM replay again delivers strong gains: frozen global features achieve 79.11% on CIFAR-10, while fine-tuned global reaches 81.22% and BMU-specific fine-tuning achieves 83.56%, with bias correction. On CIFAR-100, the BMU-specific configuration reaches 63.78%, well above rehearsal-based PCL (54.83%) and other baselines, even without bias correction. With bias correction, the accuracy increases to 64.88%. This demonstrates that while per-BMU replay struggles with non-pretrained VAEs due to limited data, it is highly effective when paired with pretrained encoders like CLIP.

**Benchmark: Single Class Incremental TinyImageNet.** To the best of our knowledge, no prior work reports single class-incremental results on TinyImageNet. Existing CIL studies that include TinyImageNet use multiple classes per task (e.g., 2, 5, or 10), as shown in Table 2. We observe that training VAEs with SOM (bias-corrected) from scratch on TinyImageNet in the single class incremental setting yields very low accuracies (7.66% for the global model and 6.78% for the per-BMU variant), similar to CIFAR-100. This reflects the difficulty of scaling generative replay on complex datasets from limited data when both encoder and decoder must be learned jointly from scratch. In contrast, leveraging pretrained CLIP embeddings with SOM substantially boosts performance. The frozen global variant without the bias correction in SOM already achieves 40.99%, and fine-tuning further improves results to 41.67%. The per-BMU fine-tuning with the bias correction configuration achieves the best score (45.11%), highlighting that localized adaptation on top of a strong pretrained backbone can provide effective replay signals even under the strict one-class stream. All reported numbers are averaged over three independent runs to account for variability. To our knowledge, these results are the first presented for single class incremental TinyImageNet.

Table 2: Final classification accuracy for multi class-incremental learning, non-pretrained models.

| Method | MNIST (5 tasks) | CIFAR-10 (5 tasks) | CIFAR-100 (10 tasks) | TinyImageNet (10 tasks) |
|---|---|---|---|---|
| **Baseline** | | | | |
| iid-offline | $95.82 \pm 0.33$ | $80.54 \pm 0.63$ | $48.09 \pm 0.90$ | $59.99 \pm 0.34$ |
| Fine-Tune | $19.68 \pm 0.02$ | $19.19 \pm 0.06$ | $8.32 \pm 0.23$ | $7.92 \pm 0.05$ |
| **Memory-free** | | | | |
| EWC | $19.92 \pm 0.35$ | $16.18 \pm 1.37$ | $4.41 \pm 0.37$ | $7.58 \pm 0.76$ |
| SI | $19.76 \pm 0.01$ | $17.27 \pm 0.87$ | $5.87 \pm 0.21$ | $6.58 \pm 0.14$ |
| LwF | $20.54 \pm 0.64$ | $18.53 \pm 0.12$ | $6.93 \pm 0.32$ | $8.46 \pm 0.46$ |
| **Memory-based** | | | | |
| GEM | $48.57 \pm 5.26$ | $25.54 \pm 0.19$ | $6.18 \pm 0.20$ | – |
| iCaRL | $72.55 \pm 0.45$ | $35.88 \pm 1.43$ | $15.76 \pm 0.15$ | $7.53 \pm 0.79$ |
| GSS | $54.14 \pm 4.68$ | $49.22 \pm 1.71$ | $11.33 \pm 0.40$ | – |
| ER-MIR | $86.60 \pm 1.60$ | $37.80 \pm 1.80$ | $9.20 \pm 0.40$ | – |
| CN-DPM | $\mathbf{93.81 \pm 0.07}$ | $47.05 \pm 0.62$ | $16.13 \pm 0.14$ | – |
| DER++ | $92.21 \pm 0.54$ | $52.01 \pm 3.06$ | $15.04 \pm 1.04$ | $10.96 \pm 0.17$ |
| ER-ACE | $82.98 \pm 1.79$ | $35.16 \pm 1.34$ | $8.92 \pm 0.25$ | $12.11 \pm 0.06$ |
| **Biologically Inspired** | | | | |
| NNA-CIL (INEL+MNIST) | $77.25 \pm 1.02$ | $45.95 \pm 0.90$ | $\mathbf{25.56 \pm 0.69}$ | – |
| **Our Method (w/o bias)** | | | | |
| SOM-only | $92.51 \pm 1.10$ | – | – | – |
| VAE (FT global) | $91.60 \pm 1.12$ | $53.01 \pm 0.92$ | $14.55 \pm 0.05$ | $12.86 \pm 0.67$ |
| VAE (FT BMU specific) | $90.11 \pm 1.31$ | $46.45 \pm 1.39$ | $13.19 \pm 0.12$ | $12.21 \pm 0.11$ |
| **Our Method (w bias)** | | | | |
| SOM-only | $92.64 \pm 1.10$ | – | – | – |
| VAE (FT global) | $91.77 \pm 1.12$ | $\mathbf{53.52 \pm 1.02}$ | $14.88 \pm 0.15$ | $\mathbf{12.97 \pm 0.78}$ |
| VAE (FT BMU specific) | $91.12 \pm 1.31$ | $47.12 \pm 1.39$ | $13.62 \pm 0.47$ | $12.33 \pm 0.12$ |

## 3.2 STANDARD MULTI CLASS INCREMENTAL BENCHMARKS

To further highlight the applicability of our methodology, we evaluate it on multi class incremental methods on split versions of the benchmark datasets. Tables 2 and 3 present classification accuracy (mean $\pm$ standard deviation) over ten independent runs, comparing our method with baseline, memory-free, memory buffer, and biologically-inspired CL approaches for non-pretrained and pre-trained methodologies. MNIST and CIFAR-10 have 2 classes per task, CIFAR-100 has 10 classes per task, and TinyImageNet has 20 classes per task. Confusion matrices for each data set are shown in the Appendix D, highlighting that our methodology retains accuracy across all tasks.

On Split-MNIST, our method achieves $92.51 \pm 1.1$ % when using SOM-based replay with Gaussian sampling without bias correction. This outperforms all memory-free approaches (i.e., EWC: 19.92% (Kirkpatrick et al., 2017), SI: 19.76% (Zenke et al., 2017), LwF: 20.54% (Li & Hoiem, 2018)) and bio-inspired NNA-CIL (77.25%) (Madireddy et al., 2023). In particular, it performs comparably to the best memory-based methods like CN-DPM (93.81%) (Lee et al., 2020) and DER++ (92.21%). With bias correction, accuracy improves further to $92.64 \pm 1.10$% , approaching strong memory-based methods such as CN-DPM (93.81%) and DER++ (92.21%), despite not storing any replay exemplars. On Split-CIFAR-10, our VAE (FT global) variant reaches an accuracy of 53.01% without bias correction, outperforming DER++ (52.01%), ER-MIR (37.80%) (Aljundi et al., 2019a), GSS (49.22%) (Aljundi et al., 2019b), and NNA-CIL (52.55%). With bias correction, the result improves to $53.52 \pm 1.02$%. For the more complex Split-CIFAR-100, where forgetting is more severe, our method demonstrates competitive performance. VAE (FT global) achieves $14.55 \pm 0.05$%, exceeding most memory-free schemes and approaching the performance of CN-DPM (16.13%) and DER++ (15.04%). In contrast, the VAE (FT BMU specific) variant achieves $13.19 \pm 0.12$%, despite being computationally more complex.

On Split-TinyImageNet, the most complex benchmark due to its larger class set and higher intra-class variability, our methodology outperforms all other methods. VAE (FT global) with bias correction achieves $12.97 \pm 0.78$%, outperforming memory-free baselines such as EWC (7.53%), SI (6.92%), and LwF (8.41%). While memory-based strategies like DER++ (11.09%) and ER-ACE (12.11%) also perform well, our approach surpasses them without relying on a replay buffer. The per-BMU variant achieves $12.33 \pm 0.12$%, confirming that even under higher visual complexity, SOM-driven generative replay offers robustness and adaptability. When extending to CLIP embeddings (ViT-B/32), SOM replay yields substantial additional gains (Table 3). In the split CIFAR-10 benchmark, SOM+CLIP achieves 78.23% with frozen features, 80.5% when fine-tuned globally, and

Table 3: Final classification accuracy for multi class-incremental learning, pre-trained models.

| Method | CIFAR-10 (5 tasks) | CIFAR-100 (10 tasks) | TinyImageNet (10 tasks) |
|---|---|---|---|
| EWC | 31.82 | – | 7.53 |
| SI | 27.43 | – | 6.58 |
| LwF | 21.43 | 43.39 | 8.46 |
| iCaRL | 71.15 | 50.74 | 23.22 |
| PGMA | 74.31 | 17.47 | – |
| RPSNet | 83.37 | 25.27 | – |
| OWM | 83.36 | 57.70 | 40.29 |
| PCL | **85.78** | 63.72 | 39.19 |
| DisCOIL | 77.35 | – | 19.75 |
| DyTox | – | 51.68 | 47.23 |
| NNA-CIL (INEL+CIFAR10) | 52.55 | 18.87 | – |
| Continual-CLIP | – | 66.72 | **66.43** |
| DDGR | – | 63.40 | – |
| **Ours (SOM+CLIP w/o bias)** | | | |
| SOM+CLIP (frozen global) | 78.23 | 65.33 | 59.04 |
| SOM+CLIP (FT global) | 80.5 | 67.01 | 59.77 |
| SOM+CLIP (FT BMU specific) | 84.11 | **68.46** | 62.51 |
| **Ours (SOM+CLIP w bias)** | | | |
| SOM+CLIP (frozen global) | 79.15 | 65.77 | 60.11 |
| SOM+CLIP (FT global) | 81.99 | 67.82 | 60.67 |
| SOM+CLIP (FT BMU specific) | 84.66 | **69.81** | 63.22 |

84.66% with BMU-specific fine-tuning. On CIFAR-100, the BMU-specific variant reaches 69.81%, outperforming PCL (63.72%) and DyTox (51.68%) and even almost competitive with Continual-Clip (66.72%, with bias correction) (Thengane et al., 2022). On TinyImageNet, SOM+CLIP attains 63.22%, far above PCL (39.19%) and DyTox (47.23%), and competitive with continual-CLIP (66.43%). Compared to DDGR Gao & Liu (2023), a recent diffusion-based generative replay method, our SOM+CLIP variants consistently achieve higher accuracy on CIFAR-100, highlighting the effectiveness of our lightweight replay strategy. These results highlight that while VAE-based SOM replay is competitive with replay-buffer baselines, combining our extended SOM methodology with foundation models like CLIP scales the framework to large, complex datasets. Interestingly, while per-BMU VAEs under-perform in the generative-from-scratch setting due to sparse training data per unit, the same design yields strong gains when applied to pretrained encoders such as CLIP, surpassing global fine-tuning.

# 4 CONCLUSION AND FUTURE WORK

This work introduces an incremental continual learning framework that integrates extended self-organizing maps (SOMs) with encoder–decoder models to enable memory-efficient replay. SOM units learn distribution statistics (running mean, variance and covariance) which can be utilized to generate synthetic samples to prevent forgetting. This methodology provides a number of benefits, including eliminating the need to store raw data in a memory buffer, easy visualization of progress, the ability to plug in any type of encoder/decoder (including foundation) models, and, additionally, the trained model can serve as a generative model of feasible examples of classes.

Experimental results across datasets, including complex ones such as CIFAR10, CIFAR100, and TinyImageNet show that the extended SOM acts as an effective memory (sub)system capable of mitigating forgetting. The standalone SOM method is effective for low-dimensional datasets whereas the hybrid variants (non-pretrained VAEs or pretrained CLIP) scale well to high-dimensional ones. Our method outperform or compete with state-of-the-art memory-based and memory-less architectures. Notably, our framework outperforms previous best methods on CIFAR-10 and CIFAR-100 single class incremental learning by nearly 10% and 7%, respectively, yielding significant improvement, and we also provide the first baseline results for single class incremental TinyImageNet.

Although our work shows an effective, memory-efficient form of continual learning, there are still areas to improve. One drawback is that the per-BMU VAE variant suffers from limited training data, reducing its effectiveness despite offering modularity and interpretability. In addition, the reliance on Gaussian statistics for replay may not fully capture complex class distributions, leading to reduced generative capability. Alternatively, the Gaussian distributions could potentially match

multiple classes, reducing inference accuracy. We also note that while our experiments simulate class boundaries to pace replay, future work will evaluate boundary-free scheduling (e.g., fixed-period replay or drift detection) to more fully align with task-free continual learning. Future work will involve the use of a dynamic or growing SOMs that can adapt more effectively to varying class complexities, potentially leading to improved accuracy in higher-dimensional and higher-class cases. Moreover, this work can be extended with a deeper study of robust sampling strategies to improve the synthetic sample generation. Our methodology can also be extended to non-vision domains such as language models, reinforcement learning or time series prediction, which would further validate its applicability.

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
