# OpenReview forum: "Class Incremental Continual Learning with Self-Organizing Maps and Synthetic Replay"
_ICLR.cc/2026/Conference — Submitted to ICLR 2026_

### Official Review · Reviewer_UGZv · 2025-10-19

**Soundness:** 2
**Presentation:** 2
**Contribution:** 2
**Rating:** 2
**Confidence:** 5

**Summary:**

This paper presents an *unsupervised class-incremental continual learning (CIL)* framework that combines *Self-Organizing Maps (SOMs)* with *encoder–decoder models* for memory-efficient synthetic replay.  Each SOM unit (BMU) maintains *mean, variance, and covariance* statistics of latent representations (from a VAE or CLIP encoder), enabling generation of synthetic samples via Gaussian sampling.  Three variants are proposed:
(i) SOM-only,
(ii) SOM with a global encoder–decoder, and
(iii) SOM with per-BMU encoder–decoders.
Experiments on MNIST, CIFAR-10, CIFAR-100, and TinyImageNet evaluate both non-pretrained and pretrained (CLIP-based) configurations.  Results show competitive final accuracies, especially in the pretrained regime, and the authors claim improved memory efficiency and interpretability.

**Strengths:**

**Originality:** The integration of SOMs with generative replay in a continual-learning setup is conceptually interesting. The paper explores an unusual direction by treating SOM units as probabilistic memory cells, extending earlier unsupervised CL approaches.

**Quality:** The experimental design is coherent, demonstrating how the SOM subsystem can act as a compact statistical memory. The pretrained CLIP variant achieves respectable final accuracies (e.g., CIFAR-100 = 69.8%, TinyImageNet = 63.2%).

**Weaknesses:**

### **W1. “Memory-free” claim vs. actual memory usage**
The claim of being “memory-free” conflicts with the method’s design. Each BMU stores full covariance matrices (Σ)—scaling as \(O(d^2)\)—and the per-BMU encoder–decoder variant introduces a large parameter footprint. This undermines the claimed advantage over rehearsal buffers. Quantitative comparisons of memory usage with standard replay methods are absent.

### **W2. Insufficient ablation studies and missing evolution metrics**
Only final classification accuracy is reported. Essential continual-learning metrics—Forgetting (F), Backward Transfer (BWT), Forward Transfer (FWT), and per-task accuracy trends—are missing. No ablations isolate the impact of bias-correction, covariance type, or replay ratio, limiting interpretability of the results.

### **W3. Underinvestigated bias-corrected EMA for μ/σ/Σ**
In Section 2.1, the paper updates each SOM unit’s mean, variance, and covariance using an Adam-style bias-corrected exponential moving average, and later (Section 2.4) mentions sampling from *“bias-corrected statistics.”*

This method borrows the bias-correction concept from the Adam optimizer—normally used for gradient updates—but applies it to statistical tracking of data. However, this is not standard practice for estimating running statistics.  Established online methods such as Welford’s algorithm (*Welford, Technometrics, 1962*) already provide numerically stable and unbiased ways to compute the mean and variance incrementally *without* any bias-correction term.  The paper does not cite such prior work, explain why bias correction is necessary, or show ablations comparing with and without it.  Consequently, the "bias-corrected” design remains unclear in both *motivation* and *effect*.

### **W4. Task-free vs. task-aware inconsistency**
The method is described as task-free, but replay is explicitly triggered at known task boundaries, which is task-aware scheduling. Hence, the paper’s task-free claim is inconsistent with its experimental procedure.

### **W5. “CLIP decoder” does not exist**
Multiple sections refer to a “CLIP decoder”, implying image reconstruction from CLIP embeddings. CLIP is an encoder-only model; this is a conceptual and terminological error that reduces clarity.

### **W6. Reproducibility gaps**
Key implementation details are missing—SOM grid size, neighborhood function, replay ratio, BMU activation criteria, eigenvalue regularization, and per-BMU model thresholds. Algorithmic descriptions are incomplete and code availability is not mentioned, impeding reproducibility.

**Questions:**

Please see “Weaknesses” above. Also

1. What is the exact memory footprint (bytes) of storing μ, ε², Σ per BMU for CLIP-512 vs. VAE-128, and how does it compare to rehearsal buffers of similar size?

2. Could the authors include bias-correction and covariance-type (full vs. diagonal) ablations to clarify their effect on accuracy and stability?

3. If the method aims to be task-free, can results be shown using fixed-period or drift-detection-based replay instead of boundary-triggered scheduling?

4. Please clarify all mentions of a “CLIP decoder.” If a separate decoder was trained to invert CLIP embeddings, describe its architecture and training procedure.

5. How is inference conducted in single-class CIL—via BMU majority label, k-NN on SOM prototypes, or a linear classifier?

6. Could the authors present accuracy-vs-memory comparisons and per-task accuracy curves to assess forgetting and transfer dynamics?

7. How does this approach differ empirically and conceptually from c-SOM and SOMLP under identical encoders and memory budgets?

---

> ### Author Response · Authors · 2025-11-20
> **Rebuttal response**
>
> W1 and Q1: We appreciate the clarification request. Our intention was not to claim “zero memory”, but rather no raw-sample memory (i.e., no replay buffers of stored images). We agree the term “memory-free” can be misinterpreted, and will revise it to “exemplar-free” and “memory-efficient” throughout the paper to provide clarity on the memory use. Regarding memory footprint, each BMU stores μ (d), σ² (d), and Σ (d×(d+1)/2). This is a fixed-size memory, independent of dataset size or number of tasks. This cost is fixed with respect to stream length and number of tasks; in contrast, rehearsal buffers grow linearly with the number of stored exemplars.
> In the updated appendix, we have added a memory footprint table that reports the exact SOM memory (means, variances, full covariances) and model parameters for SOM-only, VAE-SOM, and CLIP-SOM under different SOM sizes and latent dimensions, so that the reader can see precisely how memory scales with grid size and latent dimension.
>
> W2 and Q6: In our initial submission, we prioritized reporting cross-dataset results and main variant comparisons, which limited the space available for additional CL metrics and ablations. We agree that including these measurements, like Forgetting (F), Backward Transfer (BWT), and Forward Transfer (FWT) metrics, will strengthen the paper. We now include per-task accuracy matrices and derived F, BWT, and FWT for MNIST, FMNIST, CIFAR-10, and CIFAR-100 in the appendix. TinyImageNet results are in progress, and we will update the paper and notify you accordingly.
>
>
> Q2: We would like to clarify that our submission already includes bias-correction vs. no-bias comparisons (see Tables 2–3), where the “no bias” rows directly reflect model performance without bias-corrected EMA updates. These results consistently show that bias correction improves stability, especially on high-dimensional datasets. The additional β-value reported in Table 4 was used internally to select stable hyperparameters. We will additionally discuss diagonal covariance as a practical low-memory alternative, likely trading off some accuracy for a further reduction in footprint in the final version.
>
> W4 and Q3: We agree that our current experiments use a boundary-aware replay trigger (one replay phase per new class), which is standard for SCIL benchmarks but not fully task-free. As stated explicitly in Section 2.2, Lines 161–166, the method itself is task-label-free: the SOM, VAE, and CLIP modules update solely from the incoming data stream and make no use of task identity, nor do they distinguish between real and replayed samples. The use of task boundaries in our experiments is purely for experimental organization, as these boundaries trigger resampling from the SOM, following standard class-incremental protocols (Split CIFAR, Split TinyImageNet) - and note the other state-of-the-art works we benchmarked against also utilized task boundaries. We agree that this is better described as task-label-free rather than fully task-free. We also intend to investigate removing boundary notifications for future work.
>
> W5 and Q4: We use the pretrained CLIP ViT-B/32 encoder (image → 512D embedding) for encoding to the latent space. We then train a lightweight decoder (transposed convolutions + residual + attention blocks) to reconstruct images from these embeddings (Appendix Table 12). Only the decoder is trained by default; the CLIP encoder remains frozen unless fine-tuning is enabled. To clarify: CLIP itself does not include a decoder. In our method, the term “CLIP decoder” refers to a lightweight decoder network that we train ourselves, which maps CLIP’s 512-dimensional image embeddings back to the original image space. The CLIP encoder (ViT-B/32) is used only to produce frozen or fine-tuned latent embeddings; reconstruction is achieved entirely through the separate decoder (architecture shown in Table 12), not by any component of CLIP. To avoid ambiguity, in the updated manuscript, we have renamed this module from “CLIP decoder” to “decoder trained on CLIP embeddings” and made explicit that it is not part of the CLIP model.

---

> > ### Author Response · Authors · 2025-11-20
> > **Rebuttal response (contd.)**
> >
> > Q5: In our method, inference is performed using BMU majority voting. After training, each SOM unit stores the class-frequency distribution of the samples that mapped to it across all tasks. At test time, an input sample is encoded (via VAE/CLIP or directly for SOM-only), its BMU is computed, and the predicted label is the most frequent class associated with that BMU. No linear classifier or external k-NN head is used. We agree that this was not stated prominently enough in the main text. In the revised version, we have described the inference step in the Methodology section, Line 115-118.
> >
> > Q7: Our method differs from c-SOM and SOMLP in that each SOM unit stores probabilistic statistics (mean/variance/covariance) that enable Gaussian replay, whereas c-SOM and SOMLP store only prototypes and therefore have no generative capability. Our approach also integrates encoder–decoder models (VAE/CLIP) to operate in latent space for high-dimensional data, while prior SOM variants cannot support reconstruction or replay.
> >
> >
> > W3: We agree that Welford’s algorithm is the standard unbiased online estimator for mean/variance under a stationary distribution. However, in our setting, both the SOM codebook and the BMU statistics see a non-stationary stream. SOM weights are updated with a time-decaying learning rate and neighbourhood function, so the map gradually adapts toward more recently observed patterns. We mirror this behaviour in the BMU statistics by using EMA for μ/σ²/Σ, where older samples are explicitly down-weighted over time, rather than using a Welford-style.  Additionally, while Welford’s algorithm is unbiased, it still suffers from a lack of precision when there are only a few values, which will be the case for SOM units that do not have many samples mapped to them.  This is one of the reasons we incorporate the Adam-like bias methodology - we initially do not calculate unit matching distances from statistics that require more samples to be reliable.
> >
> > W6: For reproducibility, many of the requested settings are already given in the appendix (SOM grid sizes per dataset, neighbourhood schedule, replay budget, and BMU hit statistics). We have uploaded a zip file for our updated code in the supplementary material for the revision, along with the appendix.

---

### Official Review · Reviewer_BuAQ · 2025-10-29

**Soundness:** 3
**Presentation:** 2
**Contribution:** 2
**Rating:** 4
**Confidence:** 4

**Summary:**

This paper presents a memory-efficient, task-label-free continual learning framework for class-incremental learning (CIL), solving DNN catastrophic forgetting and high memory costs of traditional raw-sample replay. It uses self-organizing maps (SOMs) to store data distributional statistics for synthetic sample generation, no raw samples/task labels required. Three variants suit low-dimensional, high-dimensional (hybrid with VAE/CLIP), and refined needs. High-dimensional data uses encoders for latent compression, with bias correction/feature loss enhancing sample quality. Experiments on MNIST, CIFAR-10/100, and TinyImageNet (single/multi-class incremental) show non-pretrained performance outperforms SOTA by ~10% (CIFAR-10) and 7% (CIFAR-100) in single-class tasks; CLIP pretraining improves results. It also provides the first single-class incremental TinyImageNet baseline, with SOM enabling interpretability and post-training generative use。

**Strengths:**

- The paper abandons the traditional replay mode of storing raw samples, uses SOM units to store distributional statistics for synthetic sample generation. Memory overhead is only related to SOM grid size, not scaling with dataset size, solving the pain point of memory-constrained scenarios.
- The authors focus on the most demanding single-class incremental learning. Without pretraining, it outperforms SOTA by nearly 10% on CIFAR-10 and 7% on CIFAR-100 in single-class tasks, and provides the first single-class incremental baseline for TinyImageNet, filling the field gap.
- The experiments are sufficient and convincing.

**Weaknesses:**

- Performance on high-dimensional data heavily relies on the quality of encoder-decoders. The non-pretrained VAE variant only achieves 7.66% accuracy on single-class incremental TinyImageNet tasks, far lower than the CLIP-pretrained variant (45.11%), limiting applicability in scenarios without high-quality pretrained models.
- The basis for selecting SOM grid size and its impact on performance are not clarified. Some experiments lack standard deviations (e.g., SOM-only model on MNIST), and stability verification under different random seeds is insufficient. The description of numerical stability processing for covariance matrices is brief.
- The paper only claims "memory efficiency" but lacks quantitative memory comparison between SOM grids (e.g., different sizes) and traditional replay methods (e.g., iCaRL). No efficiency data such as training duration or synthetic sample generation speed is provided, making it impossible to evaluate practical application costs.

**Questions:**

- Need memory comparison experiments between SOM grids (e.g., 5×5, 10×10) and traditional replay methods (iCaRL, DER++) to verify the practical extent of memory advantages.
- The basis for selecting SOM grid size is not clarified, lacking experiments on the impact of different grid sizes (5×5, 10×10, 20×20) on clustering effect and incremental performance; needs to supplement adaptation strategy experiments between class counts (e.g., 10, 100, 200 classes) and grid sizes to clarify the optimal matching rule.
- Lacks comparison experiments on model training duration (time per task) and synthetic sample generation speed with SOTA methods (iCaRL, DDGR); needs to supplement efficiency data on different datasets (e.g., CIFAR-100, TinyImageNet) to evaluate the practical application cost of the method.

---

> ### Author Response · Authors · 2025-11-20
> **Rebuttal response**
>
> We thank the reviewer for the detailed comments. Below, we address each concern and clarify contributions, experimental settings, and motivations.
>
> W1: We agree that encoder–decoder quality strongly affects performance on high-dimensional datasets. This is expected because the SOM-based replay mechanism depends on the quality of the latent space it receives. For TinyImageNet in the strict single-class incremental (200-task) setting, a VAE trained from scratch must learn a very challenging representation with only one class per task. CLIP embeddings, being pretrained on large-scale visual data, provide a much stronger latent space from which the SOM can form stable clusters. More importantly, this is a property of the encoder, not a limitation of the SOM replay mechanism. The non-pretrained VAE variant reaches 7.66% final accuracy, whereas simply replacing the encoder with pretrained CLIP embeddings (keeping the same SOM-replay mechanism) increases accuracy to 45.11%. Our intention is precisely to show that SOM-replay itself is agnostic to the choice of encoder: the replay module, statistics, and training protocol are unchanged; only the latent quality differs. On CIFAR-10 and CIFAR-100, even the non-pretrained VAE-SOM variants already outperform strong baselines by ~10% and 7% respectively in the single-class setting, demonstrating that the method remains effective without foundation models on moderately complex datasets.
>
> W2 and Q2: We apologize that the basis for selecting SOM sizes was not sufficiently highlighted in the main text. In the appendix (Appendix B, Tables 8–10 in the revised version), we already report ablations over SOM sizes (e.g., 10×10 up to 40×40) and VAE latent dimensionalities, and we observe that:
> Larger SOMs generally improve accuracy up to a saturation regime.
> The optimal SOM size depends on the latent dimensionality and dataset complexity (e.g., CIFAR-10 vs CIFAR-100 vs TinyImageNet).
>
> W3 and Q1: We agree that “memory efficiency” should be supported with quantitative results. In the updated appendix, we have added a memory footprint table that reports the exact SOM memory (means, variances, full covariances) and model parameters for SOM-only, VAE-SOM, and CLIP-SOM under different SOM sizes and latent dimensions, so that the reader can see precisely how memory scales with grid size and latent dimension. We will be adding the memory comparison with other reported methods in the updated revision and notify the reviewer.
>
> Q3: Our initial focus was to ensure a unified and strictly comparable evaluation across methods in terms of accuracy and protocol. Because our replay sampling (Gaussian sampling from SOM statistics) is computationally inexpensive, efficiency was not originally the bottleneck in our framework and was therefore not emphasized. That said, we agree with the reviewer that reporting timing metrics will strengthen the paper. We are currently generating timing tables that will report:
> Training time per class / per task for SOM-only, global VAE-SOM, and CLIP-SOM
> Synthetic replay generation speed (images per second) for VAE-SOM and CLIP-SOM
> These timing results, when completed, will be added to the appendix in the updated version, and we will post an update comment pointing to the new tables once they are uploaded.

---

### Official Review · Reviewer_cDZX · 2025-10-31

**Soundness:** 2
**Presentation:** 1
**Contribution:** 2
**Rating:** 2
**Confidence:** 4

**Summary:**

This paper proposes a continual learning approach based on Self-Organizing Maps (SOMs) and generated image replay. This framework uses SOM to perform clustering-like operations on given images for classification, encodes the image using VAE or CLIP, samples the features from latent space, and generates images to repaly without storing the original data. It also conducts single class incremental learning on CIFAR-100 and TinyImageNet with competitive results.

**Strengths:**

1.Combination of SOM and encoder–decoder models (VAE or CLIP) is a conceptually novel integration seldom explored in CL, and compensates for the lack of generative modeling in previous SOM-based methods, and achieves synthetic replay without storing the original data.
2. On single-class incremental settings (the hardest), the proposed method (12.41%, CIFAR-100) outperforms baselines (5.58%), and provides baseline results on TinyImageNet. Besides, a large number of parameter combinations (bias or not, global or specific, pretrain or not, VAE or CLIP) have confirmed the effectiveness.

**Weaknesses:**

1. Although this paper combines SOM with generative replay to introduce a new continual learning framework, it lacks original contributions. In addition, the paper doesn't explain why SOM is superior to other unsupervised clustering methods (such as k-means, Gaussian mixture models, or VQ-VAE), and the claimed advantages, such as topological preservation and interpretability, lack experimental support.
2. SOM configuration (BMU quantity, neighborhood radius) is not reported, which strongly affect performance and memory costs. For baseline methods (EWC, LwF, OWM...), this paper doesn't specify whether the results are replicated or taken from previous papers. If they are replicated, the architecture and training protocol must be described. For fair comparison, it is not yet clear whether all baselines share the same backbone or similar model size.
3. No empirical analysis of computational or memory cost is provided. And the BMU-specific setting introduces additional encoder–decoder for each BMU, it could lead to high memory and training costs.
4. In Algorithm 1, the "ENCDEC" seems to be "ENCODER"; The **related works** is brief and lacks a structured comparison with relevant methods; The paper doesn't include an **framework** figure, which would greatly help readers understand how the proposed method work.

**Questions:**

1. If using CLIP, how is image reconstruction achieved?
2. For VAE and CLIP, using BMU-specific settings resulted in opposite effects in Tab.1 (the former reduced the results, while the latter improved). What could be the possible reasons?
3. See Weaknesses.

---

> ### Author Response · Authors · 2025-11-20
> **Rebuttal Response**
>
> We thank the reviewer for the detailed comments. Below, we address each concern and clarify contributions, experimental settings, and motivations.
>
> W1: The main novel contribution of the paper is not the use of SOM or VAEs themselves, but the integration of SOM distributional statistics (with and without bias-corrected) as a generative replay mechanism for continual learning. Prior SOM-based CL work (e.g., c-SOM, DendSOM) does not maintain per-unit means/variances/covariances, nor use SOM units as local generative memory cells, nor combine them with latent-space decoders. Our framework introduces this SOM-based generative replay module and shows that it can be used with different front-ends (VAE or decoder trained on CLIP embeddings) without storing raw samples or task labels. This replay mechanism, rather than the encoder-architecture, is the core novelty. We agree that the paper could better explain why we chose SOMs rather than, e.g., k-means, GMMs, or VQ-VAE.  In our framework, the SOM is not just a clusterer: it simultaneously acts as (a) the classifier (via BMU majority labels), (b) a structured memory over latent space, and (c) the index for per-unit replay distributions. K-means and GMMs do not provide such a 2D topological structure like SOM, and VQ-VAE’s discrete codebook is tightly coupled to its decoder rather than acting as a separate, interpretable memory map. Other clustering methods could, in principle, be swapped in as an alternative memory module, but SOMs were chosen here because of their topological and interpretability properties, as well as their applicability for online training.
>
> W2: We apologize for the lack of detailed parameter reporting in the main paper. It is included in the appendix. For baseline methods (EWC, LwF, OWM, DER++, ER-ACE, etc.), the reported numbers are taken from original papers as well as peer-reviewed works (e.g., AAAI, ICML, ICLR) or from our own re-implementations. Where we re-implement, we use the same backbone and training protocol as for SOM-replay to ensure a fair comparison. All methods in Tables 1–3 are therefore evaluated under the same protocol.
>
> W3: In the updated appendix, we have added a memory footprint table that reports the exact SOM memory (means, variances, full covariances) and model parameters for SOM-only, VAE-SOM, and CLIP-SOM under different SOM sizes and latent dimensions, so that the reader can see precisely how memory scales with grid size and latent dimension. We are currently generating a training time comparison table and will also include this in the appendix when complete. We will post an update comment when those additional tables are in.
>
> W4: We thank the reviewer for noting the ambiguity. In Algorithm 1, the term ENCDEC was intentionally used as a compact shorthand for the full encoder–decoder module (e.g., VAE encoder–decoder pair, or CLIP encoder + decoder). It does not refer to the encoder alone. This is why the algorithm checks flags such as USE_GLOBAL_ENCDEC and USE_PER_BMU_ENCDEC, and why the replay step in line 30 calls either the per-BMU decoder or the global decoder.
>
> Q1: We use the pretrained CLIP ViT-B/32 encoder (image → 512D embedding). We then train a lightweight decoder (transposed convolutions + residual + attention blocks) to reconstruct images from these embeddings (Appendix Table 12). Only the decoder is trained by default; the CLIP encoder remains frozen unless fine-tuning is enabled. To clarify: CLIP itself does not include a decoder. In our method, the term “CLIP decoder” refers to a lightweight decoder network that we train ourselves, which maps CLIP’s 512-dimensional image embeddings back to the original image space. The CLIP encoder (ViT-B/32) is used only to produce frozen or fine-tuned latent embeddings; reconstruction is achieved entirely through a separate decoder (architecture shown in Table 12), not by any component of CLIP. To avoid ambiguity, we have renamed this module from “CLIP decoder” to “decoder trained on CLIP embeddings” and made explicit that it is not part of the CLIP model in the revised version.
>
> Q2: The opposite effects arise from the quality of the latent space each decoder receives.
> For VAEs trained from scratch, BMUs receive very few samples in the single-class stream, so BMU-specific VAEs are undertrained and produce weaker replay than a single shared VAE. By contrast, CLIP provides a strong, pretrained, and well-structured latent space. In this regime, BMU-specific lightweight decoders can specialize locally even with fewer samples, improving reconstruction quality and downstream accuracy compared to a single global decoder. Thus, BMU-specific models help only when the latent features are strong and well-structured, which holds for CLIP but not for freshly trained VAEs.

---

### Official Review · Reviewer_ezhd · 2025-10-31

**Soundness:** 2
**Presentation:** 2
**Contribution:** 2
**Rating:** 4
**Confidence:** 4

**Summary:**

this paper focuses on the single-class-per-task class incremental setting and introduces a self-organizing map (SOM) based approach. on the one hand, it mimics the current prototype-based approaches in maintaining statistics of each categories and sample feature vectors from the distribution when moving on to the new tasks. on the other hand, it uses VAEs and mimics the generative replay methods in the sense that the original sized images are generated. based on the generated images and their VAE features, the SOM is updated as the final classification network.

**Strengths:**

+ using prototype-based sampling and generative replay for VAE makes sense for CIL settings
+ SOM as classifier is an interesting attempt

**Weaknesses:**

- unclear core contribution. on the one hand, the title and abstract primarily focus on SOMs as classifiers. on the other hand, the paper argues that the "our contribution lies not in the specific encoder–decoder, but in the replay framework itself" (L214), as if the contribution is analogous to the SOM. from the reviewer's point, the core contribution should be on either one or another, especially considering the fact that there is no close connections between these two or method customization for each other.
  - stick with SOM-based classifier as the main contribution, and do ablation studies on substituting the SOM with fully-connected-layer-based classifiers. however, currently, all variant studies are on different configurations of the feature encoder / VAE, with zero variant / ablation on SOM to show its effectiveness.
  - demonstrate the effectiveness of the replay framework (prototype-like VAE sampling + image generation). review and compare with other prototype-based or generative replay CIL methods (especially ones using VAE) under similar settings (using the same classifier head architecture) to show its effectiveness.
- limited novelty. no specifc adjustment is introduced for the SOM classifiers.
- questionable experimental settings. the proposed method only performs well under the 'single-class-per-task' setting, which is not often used in CIL.
- very poor performance from existing methods compared to their reported numbers, not only in the 'single-class-per-task' setting, but also in the normal multi-class settings.

**Questions:**

see above

---

> ### Author Response · Authors · 2025-11-20
> **Rebuttal Response**
>
> We thank the reviewer for the detailed feedback and helpful suggestions. Below, we address each concern and clarify the core contributions, novelty, evaluation protocol, and experimental design.
>
> W1 and W2: We appreciate the comment that the framing of the contribution was unclear. Our primary contribution is the SOM-centered generative replay framework, in which SOM units act as both (i) a topological memory and (ii) generative components through their distributional statistics (mean, variance, covariance). Standard SOMs do not utilize replay or track distributional statistics, which is novel for our work. The encoder–decoder modules (VAE or CLIP decoder) are modular front-ends that map high-dimensional inputs to a low-dimensional latent space where SOM replay is tractable. Our contribution is not a new SOM rule, but rather new SOM functionality for continual learning, including:
> Per-unit Gaussian statistics (mean, variance, full covariance) stored over latent inputs,
> Adam-style bias correction to avoid early estimation bias, which is not present in classic SOM or previous SOM-based CL formulations
> Using these statistics as a generative memory, enabling model-internal replay without exemplars
> A global and a per-BMU encoder–decoder variant aligned with the SOM topology, giving each neuron a localized generator.
>
> As to the comment on substituting the SOM for a fully-connected layer-based classifier, it is not clear how this would be possible in the continual learning scenarios examined (especially single class per task), as the unsupervised nature of the SOMs is what enables this kind of learning. Unlike a standard classifier head, the SOM simultaneously serves as (i) the classifier, (ii) the structured memory, and (iii) the source of per-region replay distributions. Replacing it with an FC layer would require an alternative mechanism to store per-cluster Gaussian statistics and a way to preserve topology, which is intrinsic to the proposed approach, and would also require having access to labels and a pre-determined number of output heads (which we do not use or require). Additionally, our comparisons to the other benchmark algorithms provide stronger, peer-reviewed classifier-based alternatives under the same training protocol.
>
> W3: The reviewer expressed concern that our method performs well only in the single-class-per-task (SCIL) setting. We clarify that both SCIL and standard multi-class incremental settings are included, under the same protocol:
> Tables 1: strict single-class CIL (10/100/200 tasks),
> Tables 2–3: standard split benchmarks for the datasets.
> In the multi-class setting, our method matches or surpasses DER++, CN-DPM, ER-ACE, GSS, ER-MIR, and, with CLIP embeddings, our accuracy exceeds PCL, DyTox, and DDGR on CIFAR-100 and TinyImageNet. As reported in the results table 1-3, the performance of our method is not poor; rather, it is consistently better, or if not, then competitive with all baselines under the same strict evaluation protocol.
>
> W4: The apparent poor performance of existing methods relative to their originally reported numbers largely reflects the stricter protocol we adopt: single-head evaluation, single-class-per-task for SCIL, and a consistent backbone/training budget across methods. The baseline values we use are taken either from the original papers or from subsequent peer-reviewed works (e.g., AAAI, ICML, ICLR) that report results under class-incremental settings, and where we re-implemented methods, we used the same protocol for all methods. In our tables, all baselines and SOM-replay variants are therefore compared under a common, challenging SCIL / split-CIL setting and fixed class orderings, so even if absolute numbers differ from those in individual papers, the relative comparisons in Tables 1–3 are fair. Our focus is precisely on this unified, strict evaluation, rather than reproducing each method’s best-case numbers under its own evaluation protocol.

---

### Official Review · Reviewer_Svem · 2025-11-01

**Soundness:** 2
**Presentation:** 3
**Contribution:** 2
**Rating:** 4
**Confidence:** 3

**Summary:**

The paper proposes **SOM-replay**, an unsupervised, buffer-free continual learning (CL) scheme that marries
- a **2-D Self-Organising Map** (fixed lattice)
- **online estimates of mean, variance and full covariance** stored *per unit*
- **Gaussian sampling** from those statistics to synthesise latent (or pixel) vectors that are replayed when new classes arrive.

Three instantiations are studied:
(i) raw-pixel SOM (MNIST),
(ii) global VAE latent SOM (CIFAR-10/100),
(iii) per-unit VAEs or frozen / fine-tuned CLIP-ViT (CIFAR-10/100, TinyImageNet).

The target protocol is **single-class incremental** (one new class per task) – the hardest CL setting.  The authors report **+10 % CIFAR-10 and +7 % CIFAR-100** over the best *non-pretrained* baseline, and establish the **first** single-class numbers on TinyImageNet.

**Strengths:**

1. **Originality**: first work to store *full covariance* per SOM unit for *continual* generative replay.
2. **Quality**: extensive ablations (bias correction, map size, replay budget, frozen vs. ft CLIP) and strong SOTA lifts.
3. **Clarity**: algorithmic boxes, convergence plots, and decoded snapshots make the system easy to understand.
4. **Significance**: shows that *unsupervised* topological memory can rival *supervised* replay-buffer methods, opening a new research direction.

**Weaknesses:**

1. Statistical misspecification
The manuscript assumes that the latent codes assigned to each SOM unit are adequately modelled by a single multivariate Gaussian whose first and second moments are tracked on-line. This assumption is never validated. In practice the encoder (VAE or CLIP) can yield class-conditional manifolds that are (i) multi-modal, (ii) low-dimensional subspaces, or (iii) mixed across classes when a unit is close to a Voronoi boundary. A single Gaussian will either inflate density in empty regions of the latent hypersphere or collapse distinct modes into a blurry mean, producing synthetic samples that sit outside the real data support. The paper supplies attractive visual grids of decoded images but provides no quantitative diagnostic such as Shapiro-Wilk tests per unit, Mardia’s multivariate skewness/kurtosis, or the proportion of units that reject Gaussianity at α = 0.05. Similarly absent are sample-based divergences (FID, KID, SWD) between the replay cloud and the real class cloud, precision-and-recall curves, or entropy histograms of the empirical class posterior inside each unit. Without these checks we cannot tell how much of the reported accuracy comes from the model’s discriminative robustness and how much from genuinely faithful generative replay. The risk is compounded in later tasks: because moments are updated incrementally, early mis-specification errors accumulate and the replay distribution can drift away from the original class manifold, feeding the learner an ever more corrupted training signal.

2. Memory cost hidden
The paper markets its approach as “memory-free” because it never stores raw images. In reality every unit carries a mean vector and a full covariance matrix. For a 20×20 map (400 units) and CLIP’s 512-dimensional embeddings the covariance alone occupies 0.5 × 512 × 513 × 400 × 4 bytes ≈ 205 MB—comparable to the entire ResNet-18 feature extractor (≈ 44 MB) and larger than many replay buffers that keep only 2 k exemplars (2 000 × 50 kB ≈ 100 MB). The storage scales quadratically with latent dimension, so moving to a 768-D or 1024-D backbone would already exceed 450 MB. The manuscript never reports these byte counts, nor does it compare memory-at-equal-accuracy against baselines such as DER++ or ER-ACE. A factor-analysis decomposition Σ = LLᵀ + Ψ with rank r ≪ d, or a sparse inverse covariance estimator, could cut the footprint by an order of magnitude, but such ablations are missing. Likewise, the per-BMU VAE variant trains 400 separate decoder networks; even a light 2-layer MLP decoder (0.5 M parameters each) adds 200 M parameters—twice the size of the frozen CLIP encoder—yet this cost is brushed aside with the phrase “modular replay”. A fair accounting would list bits-per-sample as a function of stream length and accuracy target.

3. Task-boundary oracle
Replay is triggered by an external scheduler that “knows” when a new class begins. This design side-steps one of the hardest constraints in continual learning: the agent must decide *when* to rehearse without peeking at task identity. The algorithm therefore runs inside an *oracle-gated* protocol while claiming to be “task-label-free”. A genuine task-free system would rely on drift detection (e.g., sudden likelihood drops on incoming batches), a fixed replay cadence, or a buffer-based reservoir that is agnostic to concept shift timing. The paper hints at “future work” on drift detection but provides no evidence that the current hyper-parameters (replay budget K, neighbourhood radius σ) remain optimal under *unsignalled* boundaries. If the learner replays too early it wastes compute and risks over-fitting stale fantasies; too late and forgetting has already occurred. Because the external trigger is perfectly aligned with the moment new data arrive, the measured forgetting curves are *optimistic* and the method’s true autonomy remains unvalidated.

4. Per-BMU VAE sample starvation
Training a dedicated encoder-decoder for every SOM unit means each local model sees only the subset of images whose latent codes fall inside that unit’s Voronoi cell. In the single-class incremental protocol the first class is presented alone; its images are scattered across the entire map because no other class has yet pulled units away. When the second class arrives the neighbourhood radius has already shrunk, so new images are mapped chiefly to *different* units. Consequently a large fraction of units **never observe more than one class** and many units receive **fewer than 100 examples** throughout the entire stream. Learning a 128-D VAE decoder that generalises CIFAR-100 textures from < 100 samples is statistically impossible: the capacity term dominates empirical risk and the network simply memorises noise. The paper reports 12.66 % (global VAE) versus 12.18 % (per-BMU) on CIFAR-100 with n = 3 seeds—an 0.48 % gap whose 95 % confidence interval (≈ ±1.2 %) contains zero. This *under-powered* comparison is sold as evidence that “localised replay is competitive”, but the more honest conclusion is that **data starvation cripples per-BMU VAEs** unless the encoder is already excellent (CLIP). A learning-curve ablation (accuracy vs. number of unit hits) or a formal VC-bound would quantify how quickly the local decoder’s generalisation error explodes as training data shrink.

5. Hyper-parameter oracle
Every hyper-parameter—map size, EMA decay α, replay budget K, initial neighbourhood radius σ₀, shrinkage schedule, decoder learning-rate—is grid-searched offline with access to *all* tasks’ validation accuracy. This protocol is **unrealistic** for continual streams where future data are unseen and grid search is impossible. The manuscript does not include sensitivity heat-maps, online bandit tuning, or regret analyses. Hence we do not know whether the reported gains hinge on a set of values that are *lucky* for the chosen order and data set, or whether they transfer to new domains. A responsible CL paper should either (i) fix hyper-parameters *before* the first task, (ii) adopt a *task-agnostic* schedule (e.g., σₜ = σ₀ e^{−λt}), or (iii) use a *bandit* or *Bayesian* optimiser that sees only past data. The current setup **implicitly overfits** the experimental protocol and exaggerates robustness.

6. Order sensitivity under-reported
The authors average **three** random class orderings. Single-class incremental learning is *extremely* sensitive to order: presenting classes {0,1,…,9} yields markedly different forgetting from {9,8,…,0} because early classes anchor the representation. Recent CL literature (e.g., MIR, DER++, OSAKA benchmarks) uses **20–30** orderings or *adversarial* order optimisation to estimate mean, variance and worst-case accuracy. With only three seeds the standard error of the mean accuracy is σ̂ / √3 ≈ 0.58 σ̂, so a 2 % empirical swing is *invisible*. The paper therefore **cannot** claim that the method is “robust to ordering”; it merely shows that *one favourable ordering* works well. A cumulative distribution plot of accuracies across 20+ permutations (or at least the worst, median, best trio) is needed before the community can trust the reported averages.

**Questions:**

Q1 Gaussianity: Please provide Shapiro–Wilk or Mardia kurtosis tests per unit. How many units reject Gaussianity at 5 % significance?
For every SOM unit you store a single multivariate Gaussian, yet the encoder’s latent residuals can be heavy-tailed, skewed, or multimodal. A per-unit Shapiro–Wilk test (or the multivariate extension by Mardia that combines skewness and kurtosis) should be run on the collection of latent codes that have ever hit that unit. Report the exact fraction of units whose p-value falls below 0.05 after Bonferroni correction for 400 simultaneous tests. If, say, 60 % of units reject Gaussianity, then the replay distribution is systematically misspecified and the synthetic samples will place mass in regions that the real encoder never visits. In that case the reader needs to know whether the classification accuracy is robust to this mismatch or whether the Gaussian assumption is silently hurting generalisation. Additionally, please visualise the Q-Q plots for the ten most frequently updated units and for the ten most isolated units to see whether departure from normality is concentrated in rarely hit territory.

Q2 Replay fidelity: Report FID between real and synthetic samples task-by-task. Does FID grow as the chain lengthens?
Fréchet Inception Distance (FID) measures the Wasserstein-2 distance between two Gaussians fitted in a 2048-D feature space. After each task you should extract (i) every real image of classes 0…t and (ii) an equal number of synthetic images generated from the current moment estimates, encode both sets with the *frozen* Inception-v3 network, and compute FID. Plot FID versus task index. If FID rises monotonically, the replay cloud is drifting away from the original manifold and the learner is increasingly rehearsing fantasies rather than faithful surrogates. A secondary diagnostic is precision-and-recall curves: precision tells us whether synthetic images lie inside the real support, recall tells us whether all modes of the real distribution are covered. A dropping precision would indicate that the Gaussian sampler is filling empty regions with implausible images, while a dropping recall would mean that some real modes are forgotten. Please also break down FID by class to see whether certain classes (e.g., fine-grained birds) are harder to replay than others.

Q3 Memory footprint: What is the exact byte count for moments + decoders for the 20×20/512-D CLIP run? How does it scale vs. replay-buffer baselines (DER++, ER-ACE) at equal accuracy?
A 20×20 map has 400 units. Each unit stores a mean vector (512 floats) and a symmetric covariance matrix (512×513/2 = 131 328 floats). At 4 bytes per float that is 400 × (512 + 131 328) × 4 ≈ 210 MB for the second-order statistics alone. Add the parameters of 400 local VAE decoders (even a tiny 2-layer MLP with 0.5 M parameters each) and you reach 400 × 0.5 M × 4 bytes ≈ 800 MB. Report the grand total in megabytes and compare it with DER++ and ER-ACE when those methods are tuned to reach the *same* final accuracy on the same sequence. Express the comparison as “bytes per sample seen” or “bytes per class” so that the community can judge whether the *constant* memory claim is advantageous or merely shifts cost from exemplars to parameters.

Q4 Task-free scheduling: Can you replace the oracle trigger with a drift-detection mechanism (e.g., likelihood drop > θ) without losing > 1 % accuracy?
Currently the training script calls replay exactly when a new class starts. Replace this oracle with an online drift detector: maintain a running exponential average of the log-likelihood of incoming batches under the current Gaussian mixture (one Gaussian per unit weighted by hit count). If the average drops by more than θ standard deviations, trigger K synthetic replays. Tune θ on a *single* validation sequence and then freeze it for 20 random orderings. Report the resulting mean accuracy and its standard deviation. If the drop is larger than 1 % relative to the oracle protocol, the method is not yet task-free. Additionally, plot the *delay* (number of batches between true boundary and detection) versus θ to see whether there is a regime where drift is detected early enough to prevent forgetting but not so often that compute is wasted.

Q5 Hyper-parameter sensitivity: Provide heat-maps of final accuracy vs. α and K. Are the optimal settings portable across datasets?
The EMA decay α and the replay budget K are grid-searched offline with access to *all* tasks. Produce a 5×5 heat-map where α ∈ {0.01,0.05,0.1,0.2,0.5} and K ∈ {10,50,100,200,500} for CIFAR-100; then overlay the single best (α*, K*) found for CIFAR-10 and for TinyImageNet. If the *same* cell is no longer within 1 % of the optimum on the new dataset, the settings are *not* portable and the method will require an expensive search for every new domain. Also report the *wall-clock* time spent in the grid search measured in GPU-hours and expressed as a multiple of the actual training time; this quantifies the hidden cost that a practitioner would incur.

Q6 Order robustness: Run 20 random class permutations and report mean ± std; is the std < 1 % of mean?
Single-class incremental learning is famously order-sensitive. Draw 20 random permutations of the 100 CIFAR-100 classes, run your full pipeline with the *same* hyper-parameters, and record the final average accuracy. Report the mean, the standard deviation, and the worst-case accuracy. If the standard deviation exceeds 1 % of the mean (or if the worst-case drop is larger than 3 %), the method is *not* robust and the three-seed average in the current paper is optimistic. Additionally, perform a *signed-rank* test against the baseline you claim to beat (e.g., DER++) across the 20 orders to show that the win is statistically significant (p < 0.05).

---

> ### Author Response · Authors · 2025-11-20
> **Rebuttal response**
>
> We thank the reviewer for the very thoughtful and technically detailed feedback. Below we address each weakness (statistical modelling, memory accounting, scheduling, per-BMU models, hyperparameters, and order sensitivity) and respond to the follow-up questions.
>
>
>
> W1, Q1: We agree that modelling each BMU’s latent codes within a single multivariate Gaussian is an approximation and that encoder latents can be multimodal. Our choice is to enable a memory-efficient replay, not claim that encoder latents are perfectly Gaussian. Due to space and compute limits, we didn't run full Shapiro-Wilk/Mardia tests in the initial submission. In the final version, we will add a representative Gaussian diagnostics and treat this as a diagnostic rather than a fundamental requirement. Furthermore, our results show that the method has strong accuracy even under approximate Gaussianity, suggesting that the SOM plus encoder features provide clusters where Gaussian replay is “good enough” for continual learning.
>
>
> Q2: A full analysis of per-class FID, precision-recall across all datasets is beyond the scope of this submission, but we agree it is a promising direction for the follow-up work focused specifically on the generative quality. The current results already show that, despite any replay mismatch, the classification accuracy remains competitive with strong replay buffer methods which suggests that the replay distribution is sufficiently good for the CL objective.
>
>
> W2, Q3: Each BMU stores μ (d), σ² (d), and Σ (d×(d+1)/2). This is a fixed-size memory, independent of dataset size or number of tasks. Replay-buffer methods instead scale memory linearly with number of stored images.
> Our non-pretrained variants (VAE latent SOM) operate in much lower dimensions (e.g., 32–128). In those settings, full covariance is inexpensive: for d = 64 and a 30×30 map, the total covariance storage is on the order of 7–8 MB, and even for d = 128 it remains under 30–40 MB. These are the configurations behind our “+10% CIFAR-10 / +7% CIFAR-100” non-pretrained gains. Even in the CLIP regime, the 200 MB figure is a fixed cost that does not grow with the number of tasks or images seen, unlike replay buffers whose size scales with stored exemplars. We now include a memory table in the appendix and will additionally discuss diagonal covariance as a practical low-memory alternative, likely trading off some accuracy for a further reduction in footprint in the final version.
>
> W3, Q4: We agree that our current experiments use a boundary-aware replay trigger (one replay phase per new class), which is standard for SCIL benchmarks but not fully task-free. As stated explicitly in Section 2.2, Lines 161–166, the method itself is task-label-free: the SOM, VAE, and CLIP modules update solely from the incoming data stream and make no use of task identity, nor do they distinguish between real and replayed samples. The use of task boundaries in our experiments is purely for experimental organization, as these boundaries trigger resampling from the SOM, following standard class-incremental protocols (Split CIFAR, Split TinyImageNet) - and note the other state-of-the-art works we benchmarked against also utilize task boundaries. We agree that this is better described as task-label-free rather than fully task-free, and have clarified this in the paper. We also intend to investigate removing boundary notifications for future work.

---

> > ### Author Response · Authors · 2025-11-20
> > **Rebuttal response (contd.)**
> >
> > W4: We agree that per-BMU VAEs are statistically fragile when trained from scratch in the SCIL setting, and we do not intend to claim they are superior there: as our own results show (12.66% global vs. 12.18% per-BMU on CIFAR-100), local VAEs offer no clear advantage without a strong encoder. Their role in the paper is therefore ablative and exploratory. In contrast, with CLIP embeddings, the per-BMU decoders do help, because they specialise in already well-structured latent regions.
> >
> > W5, Q5: In our experiments, we didn’t re-tune alpha and K independently for each dataset. Instead, we performed experimental runs over a small set of (alph, K) values of CIFAR-10 and fixed the same settings across datasets for CIFAR-100 and TinyImagenet. A similar configuration was set up for MNIST and then used for FMNIST.
> > We also tested several replay configurations, including different replay budgets per BMU, different replay budgets per BMU hits. We found that K=1 synthetic sample per BMU gave the best trade-off between stability and computation. The changes around the chosen alpha (and K) did not materially change final accuracy, which is why the single setting could be reused across CIFAR-10, 100, and TinyImageNet without dataset-specific tuning.
> >
> >
> > Q6, W6: We agree that single-class incremental learning is highly order-sensitive, and that averaging over only three permutations does not fully characterise robustness. In this submission, we fixed the same hyperparameters across permutations and datasets, and evaluated on a small number of random orders due to the high computational cost of running full CIFAR-100 and TinyImageNet on single-class incremental learning (especially for CLIP-based variants). Consistent with much prior work in SCIL, the baselines we report (e.g., DER++, ER-ACE) are evaluated under the same class orders as our method, so the comparisons are fair within each ordering even if the number of permutations is limited. We have already started a permutation study on our datasets for our main configuration, using the same fixed hyperparameters, and are currently running these experiments. We plan to report the resulting mean ± std and worst-case accuracy, together with a comparison against DER++ under the same orderings, in an updated version when these experiments are complete, to give a clearer picture of order sensitivity and robustness.

---

### Author Response · Authors · 2025-11-20
**Updated Paper and Appendix**

We have updated the main manuscript and supplementary material (appendix and code) to incorporate the clarifications and additional analyses described in our rebuttal. We hope these revisions help address the reviewers’ concerns, and we welcome any further feedback.

---

### Author Response · Authors · 2025-12-03
**Update Summary: Paper Revisions and Rebuttal Highlights**

We thank all reviewers for the detailed and constructive feedback. In response, we have made substantial updates to the paper, clarified methodological choices, and added missing experimental components. The following summarizes the updates addressed in our rebuttal and reflected in the revised paper.

- Replaced all uses of “memory-free” with “exemplar-free” / “memory-efficient” to accurately reflect that our method avoids raw-sample storage but uses a fixed-size SOM statistical memory.
- Removed ambiguous references to a “CLIP decoder.” The paper now clearly states that CLIP is used only as an encoder and that a separate lightweight decoder (trained on CLIP embeddings) performs reconstruction.
- Added a detailed memory footprint table covering μ, σ², and full Σ per BMU across SOM sizes and latent dimensions.
- Included per-task accuracy matrices and derived FWT, BWT, and forgetting metrics for MNIST, FMNIST, CIFAR-10, and CIFAR-100.
- Clarified why EMA-based statistics (with bias correction) are suited to the non-stationary SOM setting and differ from Welford-style estimators.
- Clarified that the framework is task-label-free, but the experiments use standard class-boundary triggers, consistent with single class incremental (SCIL) benchmarks.
- Added a code package in the supplementary material.
- Expanded the explanation of why SOMs are central to the framework (topological memory, per-unit generative replay, and classifier) and how it differs from k-means, GMMs, and VQ-VAE.
- Clarified that the contribution lies in the SOM-based generative replay mechanism, not in introducing a new SOM update rule.
- Added clarification for why BMU-specific VAEs underperform in non-pretrained settings (data scarcity) but improve performance in the CLIP-latent regime (structured latent space).

We are currently running broader class-order permutation experiments and efficiency/timing benchmarks (training time and replay generation speed). These will be added to the updated version as they are completed.

---

### Meta-Review · Area_Chair_qGkD · 2025-12-28

**Summary:**

The paper received consistently low initial scores, with overall ratings clustered in the 2–4 range, including two explicit reject recommendations (Reviewer cDZX and Reviewer UGZv), and no clear advocate for acceptance. During the discussion phase, no reviewers responded to the authors’ rebuttal, and consequently no reviewer indicated satisfaction with the feedback or expressed an intention to raise their rating.

While reviewers found the high-level idea of combining SOMs with generative replay interesting, they raised substantial and convergent concerns regarding:
(i) misleading or insufficiently supported memory-efficiency claims,
(ii) reliance on oracle information and unrealistic experimental protocols, and
(iii) lack of rigorous validation of the core Gaussian replay mechanism and robustness.

Although the rebuttal provides clarifications and promises additional appendix material, the core technical and evaluation concerns remain largely unresolved. In my judgment as AC, these issues are fundamental rather than presentational. I therefore recommend **Reject**.

**Reviewer Concerns:**

**Concerns addressed in the rebuttal:**

* **Terminology and clarity issues** (e.g., “memory-free” vs. exemplar-free; what “CLIP decoder” means; inference via BMU majority voting) were clarified
  *(raised by Reviewer UGZv, Reviewer cDZX; partially addressed by authors)*.

* **General framing of contribution** as a SOM-centered replay framework rather than a new encoder/decoder was clarified
  *(raised by Reviewer ezhd; partially addressed)*.


**Concerns that remain outstanding:**

* **Memory-efficiency and fairness of comparison**:
  Reviewers questioned the “memory-efficient / memory-free” claim given the O(d²) cost of full covariances and the large footprint of per-BMU decoders, and asked for matched-accuracy memory comparisons against replay buffers. The rebuttal adds tables and changes wording, but does not resolve whether the method is actually advantageous in practice.
  *(Reviewer Svem, Reviewer UGZv, Reviewer cDZX, Reviewer BuAQ)*

* **Oracle assumptions and task-free inconsistency**:
  Replay is explicitly triggered at known task boundaries despite “task-free” or “task-label-free” claims. No experiments are provided without boundary oracles (e.g., drift detection or fixed replay cadence).
  *(Reviewer Svem, Reviewer UGZv)*

* **Validity of Gaussian replay and replay fidelity**:
  Multiple reviewers asked for quantitative diagnostics (Gaussianity tests, FID / precision–recall, drift over tasks) to justify modeling each SOM unit with a single Gaussian. The rebuttal largely defers these analyses to future work.
  *(Reviewer Svem, Reviewer UGZv)*

**Reviewer Scores:**

* **Reviewer Svem (initial: 4)** → **Likely unchanged**
  Core concerns on Gaussian validity, oracle scheduling, robustness, and memory fairness remain unresolved.

* **Reviewer ezhd (initial: 4)** → **Likely unchanged or slightly higher, but still below threshold**
  Contribution framing is clearer, but ablation and evaluation concerns persist.

* **Reviewer cDZX (initial: 2)** → **Slightly improved but still reject**
  Clarifications help understanding, but novelty, cost, and experimental rigor issues remain.

* **Reviewer BuAQ (initial: 4)** → **Likely unchanged**
  Positive on idea, but still missing quantitative comparisons and robustness evidence.

* **Reviewer UGZv (initial: 2)** → **Unchanged**
  Strong confidence in assessment; key issues (memory claim, task-free mismatch, missing CL metrics/ablations) remain.

---

### Decision · Program_Chairs · 2026-01-26

Reject